# More Time-Space Tradeoffs for Finding a Shortest Unique Substring

**Hideo Bannai** [1] ⬤**, Travis Gagie** [2] ⬤**, Gary Hoppenworth** [3],*****, Simon J. Puglisi** [4]
**and Luís M. S. Russo** [5] ⬤

[1]  M&D Data Science Center, Tokyo Medical and Dental University, Tokyo 113-8510, Japan;
   hdbn.dsc@tmd.ac.jp
[2]  Faculty of Computer Science, Dalhousie University, Halifax, NS B3H 4R2, Canada; travis.gagie@dal.ca
[3]  Department of Computer Science and Mathematics, University of Central Florida, Orlando, FL 32816, USA
[4]  Department of Computer Science, University of Helsinki, 00100 Helsinki, Finland; puglisi@cs.helsinki.fi
[5]  INESC-ID and Instituto Superior Técnico, Universidade de Lisboa, 1649-013 Lisboa, Portugal;
   luis.russo@ist.utl.pt
*****  Correspondence: garyhoppenworth@knights.ucf.edu

**Abstract:** We extend recent results regarding finding shortest unique substrings (SUSs) to obtain new time-space tradeoffs for this problem and the generalization of finding $k$-mismatch SUSs. Our new results include the first algorithm for finding a $k$-mismatch SUS in sublinear space, which we obtain by extending an algorithm by Senanayaka (2019) and combining it with a result on sketching by Gawrychowski and Starikovskaya (2019). We first describe how, given a text $T$ of length $n$ and $m$ words of workspace, with high probability we can find an SUS of length $L$ in $O(n(L/m)\log L)$ time using random access to $T$, or in $O(n(L/m)\log^2(L)\log\log\sigma)$ time using $O((L/m)\log^2 L)$ sequential passes over $T$. We then describe how, for constant $k$, with high probability, we can find a $k$-mismatch SUS in $O(n^{1+\epsilon}L/m)$ time using $O(n^\epsilon L/m)$ sequential passes over $T$, again using only $m$ words of workspace. Finally, we also describe a deterministic algorithm that takes $O(n\tau\log\sigma\log n)$ time to find an SUS using $O(n/\tau)$ words of workspace, where $\tau$ is a parameter.

**Keywords:** shortest unique substring; k-mismatch SUS; time-space tradeoff; Karp–Rabin; sketching; suffix trees

## 1. Introduction

A shortest unique substring (SUS) of a given text $T[1..n]$ is a substring containing a given position $T[q]$ and occurring only once in $T$, such that every shorter substring containing $T[q]$ occurs at least twice in $T$. For example, if $T[1..11] = $ ABRACADABRA and $q = 3$, then $T[3..5] = $ RAC is an SUS: it contains $T[3]$, it occurs only once in $T$, and each of the shorter substrings containing $T[3]$—i.e., $T[3] = $ R, $T[2..3] = $ BR and $T[3..4] = $ RA—occurs at least twice in $T$. The problem of finding SUSs has attracted significant attention recently and many variants have been proposed—interval-SUSs, $k$-mismatch SUSs, palindromic-SUSs and range-SUSs—so what we refer to simply as SUS here are sometimes also called position-SUSs. In this paper, we are only interested in position-SUSs and $k$-mismatch SUSs, which we describe shortly. We refer readers to Abedin et al.'s very recent survey [1] for a more detailed discussion.

Finding exact and approximate SUS has several applications in bioinformatics, including alignment-free genome comparison, PCR primer design, and the identification of DNA signatures distinguishing closely related organisms [2–4]. Pei, Wu, and Yeh [3] gave the definition of SUSs above, along with an $O(n)$-time and $O(n)$-space algorithm for finding an SUS for $T[q]$ given $q$,

and an $O(n^2)$-time, $O(n)$-space construction algorithm for an $O(n)$-space data structure that in $O(1)$ time returns the endpoints of an SUS for $T[q]$ given $q$. Hu, Pei, and Tao [5], İleri, Külekci and Xu [6], and Tsuruta, Inenaga, Bannai, and Takeda [7] independently improved the construction time to $O(n)$. Belazzougui and Cunial [8] reduced both the construction space and the space of the data structure to $O\left(\frac{n \log \sigma}{\log n}\right)$ at the cost of increasing the construction time to $O(n \log \sigma)$, where $\sigma$ is the size of the alphabet, while keeping the query time constant. Ganguly et al. [9] gave the following time-space tradeoffs for finding SUSs, assuming we have random access to $T$ and $\tau$ is a parameter:

- given $q$, we can find an SUS for $T[q]$ in $O(n\tau^2 \log(n/\tau))$ time and $O(n/\tau)$ space;
- in $O(n\tau^2 \log n)$ time and $O(n/\tau)$ words plus $4n + o(n)$ bits of space, we can build a $(4n + o(n))$-bit data structure answering SUS queries in $O(1)$ time; and,
- we can change the running time in both cases to $O(n\tau \log^{c+1} n)$ at the cost of increasing the space by an additive $O(n/\log^c n)$ and allowing for a low probability that the substrings are not shortest.

Recently, Senanayaka [10] gave a simple low-memory randomized algorithm based on Karp–Rabin pattern matching, but did not give theoretical bounds for it.

Hon, Thankachan, and Xu [11] generalized the problem by defining a $k$-mismatch SUS to be a shortest substring containing $T[q]$ that is not only unique, but also not within Hamming distance $k$ of any other substring of $T$. For example, if $T[1..11] = \text{ABRACADABRA}$ and $q = 5$, then $T[3..5] = \text{RAC}$ is a one-mismatch SUS, because no other substring is within Hamming distance 1 of it and each of the shorter substrings containing $T[5]$—i.e., $T[5] = \text{C}$, $T[4..5] = \text{AC}$ and $T[5..6] = \text{CA}$—is within edit distance of some other substring. On the other hand, $T[4..6] = \text{ACA}$ is not a 1-mismatch SUS, although it also has length 3, because the Hamming distance between ACA and ADA is 1.

Hon et al. gave an $O(n^2)$-time, $O(n)$-space construction of an $O(n)$-space data structure, which, given $q$, in $O(1)$ time returns the endpoints of a $k$-mismatch SUS for $T[q]$. Allen, Thankachan, and Xu [12] reduced the construction time to $O(n \log^k n)$ at the cost of increasing the construction space to $O(kn)$, and Schultz and Xu [13] gave a GPU algorithm that is fast in practice.

In Section 2, we show how Senanayaka's approach can be extended, such that, given $q$ and $m$ words of workspace and random access to $T$, with high probability we can find an SUS containing $T[q]$ in $O((L/m + 1)n \log L)$ time. In Section 3, we show that, replacing Karp–Rabin pattern matching by a result on sketching by Golan and Porat [14], we can use $O((L/m + 1) \log^2 L)$ sequential passes over $T$ instead of random access, at the cost of increasing the time to $O((L/m + 1)n \log^2 L)$ and requiring $m = \omega(\log L)$. Replacing Golan and Porat's result by one by Gawrychowski and Starikovskaya [15], for constant $k$, we can find a $k$-mismatch SUS for $T[q]$ in $O(n^{1+\epsilon} L/m)$ time using $O(n^\epsilon L/m)$ sequential passes over $T$, now requiring $m = \omega(n^\epsilon)$. Although the sketching results that we rely on are too sophisticated for us to explain them here, and to recent for us to be able to refer to one other than Golan and Porat's and Gawrychowski and Starikovskaya's papers themselves, we only use them only as black boxes, without relying on the details of how they work.

In Section 4, we describe a deterministic algorithm that makes use of directed acyclic word graphs (DAWGs) [16], the Crochemore-Perrin pattern matching algorithm [17], and suffix trees [18–20], to compute an SUS in $O(n\tau \log \sigma \log n)$ time using $O(n/\tau)$ words of workspace, improving Ganguly et al.'s result when $\tau \log(n/\tau) = \omega(\log \sigma \log n)$. Finally, in Section 5 we discuss some possible directions for future work. Table 1 summarizes known bounds for finding SUSs and $k$-mismatch SUSs, including those that we give in this paper.

**Table 1.** Previous bounds and our results for finding SUSs (the first twelve results) and $k$-mismatch SUSs (the last three), where $n$ is the length of the text, $\sigma$ is the alphabet size, $L$ is the length of the SUS or $k$-mismatch SUS, and $m, \tau < n$ and $c$ are parameters. Theorem 3 is Monte Carlo and we require $m = \omega(n^{\epsilon})$, so that the probability of failure can be made inversely proportional to any fixed polynomial of $n$. All of the data structures return an SUSs given $q$ in $O(1)$ time and have the same final space as construction space, except that the construction-space bounds for Ganguly et al.'s third and fourth results are $O(n/\tau)$ words plus $4n + o(n)$ bits of space, and $O(n/\tau + n/\log^c n)$ words plus $4n + o(n)$ bits.

| Source | Time | Space (Words) | Deterministic | Data Structure |
|---|---|---|---|---|
| Pei et al. [3] | $O(n)$ | $O(n)$ | yes | no |
| | $O(n^2)$ | $O(n)$ | yes | yes |
| Hu et al. [5] | $O(n)$ | $O(n)$ | yes | yes |
| İleri et al. [6] | $O(n)$ | $O(n)$ | yes | yes |
| Tsuruta et al. [7] | $O(n)$ | $O(n)$ | yes | yes |
| Belazzougui and Cunial [8] | $O(n \log \sigma)$ | $O(n \log(\sigma)/\log n)$ | yes | yes |
| Ganguly et al. [9] | $O(n\tau^2 \log(n/\tau))$ | $O(n/\tau)$ | yes | no |
| | $O(n\tau \log^{c+1} n)$ | $O(n/\tau + n/\log^c n)$ | no | no |
| | $O(n\tau^2 \log n)$ | $4n + o(n)$ bits | yes | yes |
| | $O(n\tau \log^{c+1} n)$ | $4n + o(n)$ bits | no | yes |
| Theorem 1 | $O(n(L/m) \log L)$ | $m$ | no | no |
| Theorem 4 | $O(n\tau \log \sigma \log n)$ | $O(n/\tau)$ | yes | no |
| Hon et al. [11] | $O(n^2)$ | $O(n)$ | yes | yes |
| Allen et al. [12] | $O(n \log^k n)$ | $O(kn)$ | yes | yes |
| Theorem 3 | $O(n^{1+\epsilon} L/m)$ | $m$ | no | no |

## 2. Tradeoffs with Karp-Rabin Pattern Matching

If we know an SUS for $T[q]$ has length at most $L$, then we can search in $T$ for repetitions of the substrings of $T[q - L + 1..q + L - 1]$ in $T$, using $O(L)$ words of workspace and $O(n)$ time. To do this, we build a suffix tree [18–20] for $T[q - L + 1..q + L - 1]$ and scan $T$, always descending in the suffix tree as much as possible and then following a suffix link. Suppose that, at some point, we have just read $T[i]$ and we are at string depth $d$ in the suffix tree. If $T[i - d + 1..i]$ is not completely contained in $T[q - L + 1..q + L - 1]$ and $d$ is the largest string depth we have reached along the edge we are currently descending, then we mark with $d$ the node below that edge. After we have scanned $T$, we can extract from the marked suffix tree the shortest string, such that:

- its locus is a leaf (meaning it occurs only once in $T[q - L + 1..q + L - 1]$),
- that leaf's label is at most $q$ and its label plus the string's length is at least $q$ (so an occurrence of the string contains $T[q]$), and
- that leaf is not marked with a number greater than or equal to the strings length (meaning we have not seen a copy of the string elsewhere in $T$).

If we do not know $L$, then we can find it via exponential search, still using $O(L)$ words of workspace, but $O(n \log L)$ time and $O(\log L)$ sequential passes over $T$. Since finding an SUS is relatively easy if we can use workspace proportional to its length, even if that length is unknown, in this paper we assume that we have less workspace.

Obviously, we can find an SUS containing $T[q]$ while using $O(1)$ words of workspace if we are willing to spend $O(L^3 n)$ time, where $L$ is the length of that SUS. For example, we can use the simple Algorithm 1. This algorithm can easily be improved to take $O(L^2 n \log L)$ time by replacing the linear search for $L$ with an exponential search. It can then be further improved to take $O(Ln \log L)$ time with high probability, while still using $O(1)$ words of workspace, by replacing naïve pattern matching with Karp–Rabin pattern matching. The resulting randomized algorithm can be either Monte Carlo if we do not verify matches or Las Vegas if we do, and the probability of failure can be made inversely proportional to any fixed polynomial of $n$ without changing the asymptotic bounds.

---

**Algorithm 1** An $O(L^3 n)$-time algorithm to find an shortest unique substring (SUS) of $T[1..n]$ containing $T[q]$, where $L$ is the length of that SUS.

---

```
 1   for ℓ from 1 to n
 2   % check each length ℓ in increasing order
 3       for i from q − ℓ + 1 to q
 4       % check all substrings of length ℓ that include T[q]
 5           unmatchedflag ← true
 6           % we have not yet seen a repetition of T[i..i + ℓ − 1]
 7           for j from 1 to n − ℓ + 1
 8               if j ≠ i
 9               % for j ≠ i, compare T[j..j + ℓ − 1] to T[i..i + ℓ − 1]
10                   matchflag ← true
11                   % we have not yet seen a mismatch between T[j..j + ℓ − 1] and T[i..i + ℓ − 1]
12                   for k from 0 to ℓ − 1
13                       if T[j + k] ≠ T[i + k]
14                       % we have found a mismatch
15                           matchflag ← false
16                           break
17                       end if
18                   end for
19                   if matchflag = true
20                   % we have found a repetition of T[i..i + ℓ − 1]
21                       unmatchedflag ← false
22                       break
23                   end if
24               end if
25           end for
26           if unmatchedflag = true
27           % there is no repetition of T[i..i + ℓ − 1]
28               print i, ℓ
29               return
30               % do not check longer substrings
31           end if
32       end for
33   end for
```

---

If we allow ourselves $m$ words of workspace, then we can make the algorithm run in $O((L/m + 1)n \log L)$ time with high probability. To do this, when searching for repetitions of the $\ell$ substrings of length $\ell$ that contain $T[q]$, we process them in $\lceil \ell/m' \rceil$ batches of size $m'$, where $m' = \Theta(m)$ depends on the ratio of the word-size to $\log_2 n$ and the power of $n$ we want in the denominator in the probability of failure. We note that we compute the hashes of the substrings in the same batch by rolling them, in $O(L)$ total time, rather than computing each of them from scratch, which would take $O(L^2)$ total time.

**Theorem 1.** *With $m$ words of workspace, with high probability we can find an SUS for $T[q]$ in $O(n(L/m) \log L)$ time.*

## 3. Tradeoffs with Sketching

For Karp–Rabin pattern matching, we must keep track of characters leaving a sliding window, for which we need either enough memory to store the contents of the sliding window—which, by assumption, we do not have—or random access to $T$. However, Golan and Porat [14] gave a Monte-Carlo randomized sketching algorithm that takes $d$ patterns of maximum length $\ell$, scans $T$ one character at a time, and, for each position, reports the longest pattern with an occurrence ending at that position with probability of failure inversely proportional to any fixed polynomial in $n$. Their algorithm uses $O(\log \log \sigma)$ time per character of $T$ and $O(d \log \ell)$ space and does not use a sliding window, so it needs only sequential access to $T$. Replacing Karp–Rabin pattern matching with Golan and Porat's

result and searching for substrings of length $\ell$ in batches of size $\Theta(m/\log\ell)$, so that we stay within our workspace bound $m$, we obtain the following result:

**Theorem 2.** *With $m = \omega(\log L)$ words of workspace, with high probability we can find an SUS for $T[q]$ in $O(n(L/m)\log^2(L)\log\log\sigma)$ time using $O((L/m)\log^2 L)$ sequential passes over $T$.*

Because Golan and Porat's algorithm is Monte Carlo, so is our result; unlike Theorem 1, we cannot easily make it Las Vegas, since verifying matches requires random access. Again, our batch size depends on the ratio of the word-size to $\log_2 n$ and the power of $n$ we want in the denominator in the probability of failure. The requirement that $m = \omega(\log L)$ means the probability of failure can still be made inversely proportional to any fixed polynomial of $n$.

Gawrychowski and Starikovskaya [15] considered a harder version of the problem Golan and Porat studied, in which we are given a distance $k$ and, for each position in $T$, we should report all of the patterns within Hamming distance $k$ of substrings of $T$ ending at that position. They gave a Monte-Carlo randomized sketching algorithm that searches for $d$ patterns of length at most $\ell$ using $O(k\log^k d\,\text{polylog}\,(\ell) + \text{occ})$ time per character of $T$, where occ is the number of matches reported ending at that character, and $O(kd\log^k d\,\text{polylog}\,(\ell))$ space. Replacing Golan and Porat's algorithm with Gawrychowski and Starikovskaya's and searching for substrings of length $\ell$ in batches of size $\Theta(m/n^\epsilon)$ for some positive constant $\epsilon$, so that we stay within our workspace bound $m$ for constant $k$, we obtain the following result:

**Theorem 3.** *For constant $k$ and with $m = \omega(n^\epsilon)$ words of workspace, with high probability we can find a $k$-mismatch SUS for $T[q]$ in $O(n^{1+\epsilon}L/m)$ time using $O(n^\epsilon L/m)$ sequential passes over $T$.*

Like Theorem 2, this result is Monte Carlo and the probability of failure is inversely proportional to any fixed polynomial in $n$.

## 4. A Deterministic Algorithm

**Lemma 1.** *Given a length $\ell$ and a position $p$ in a text $T[1..n]$ with alphabet size $\sigma$, there exists a deterministic algorithm that can determine whether there is a substring of length $\ell$ that covers position $p$ and is unique in $T$ in $\mathcal{O}((\frac{\ell}{m}+1)n\log\sigma)$ time and $\mathcal{O}(m)$ words of workspace.*

**Proof of Lemma 1.** There are at most $\ell$ possible substrings of length $\ell$ covering $p$ that could be unique. We separate these substrings into $\lceil\frac{\ell}{m}\rceil$ batches of $m$ substrings with adjacent starting positions. Note that there may be one remainder batch with less than $m$ substrings.

Consider a batch $B$ containing $k$ substrings of length $\ell$, where $1 \leq k \leq m$. Let $B_i$ denote the substring in $B$ with the $i$th adjacent starting position in $B$. Subsequently, $B_1$ is the leftmost substring in $B$ and $B_k$ is the rightmost substring in $B$.

Consider the following substrings:

$x = B_1[1..k-1]$

$z = B_k[\ell-k+2..\ell]$

$y = B_1[k..\ell] = B_k[1..\ell-k+1]$

It is useful to think of $x$ as the prefix of $B_1$ that does not overlap with $B_k$ in $T$. Similarly, $z$ is the suffix of $B_k$ that does not overlap with $B_1$. Finally, $y$ is the suffix of $B_1$ and the prefix of $B_k$ that overlap with each other in $T$. Note that any substring in $B$ is equal to $x_{suff} \cdot y \cdot z_{pref}$, where $x_{suff}$ is some suffix of $x$, $z_{pref}$ is some prefix of $z$ and $\cdot$ denotes concatenation. For each batch, we will use the substrings $x$, $y$, and $z$ to determine whether any of the $k$ substrings in the batch occur in $T$.

We can scan $T$ to enumerate all occurrences of suffixes of $x$, all occurrences of $y$, and all occurrences of prefixes of $z$. Using this information we can determine if any of the $k$ substrings in the batch occur elsewhere in $T$ and are therefore not unique. Suppose that we search text $T$ and we find an occurrence of a suffix $q$ of $x$, an occurrence $r$ of $y$, and an occurrence of a prefix $s$ of $z$. If $q = T[i..j]$,

$r = T[j+1..j+1+\ell-k]$ and $s = T[j+2+\ell-k..\ell+i-1]$ for some $i, j \in [1, n], i \leq j$, then we have found an occurrence of a substring in batch $B$ in $T$. Specifically, if $qrs = B_a$ for some $B_a \in B$, then $B_a$ is not unique in $T$. $\square$

### 4.1. Finding Occurrences of Suffixes of X

To enumerate all occurrences of suffixes of $x$ in $T$, we construct the Directed Acyclic Word Graph (DAWG) [16] $D(x\$)$ with suffix links. $D(x\$)$ is the smallest deterministic automaton which accepts all suffixes of $x\$$ and it is known to have the following properties:

1.  Each edge is labeled by a single symbol, and the labels of all outgoing edges from a given node are distinct. The total number of nodes and edges is linear in the length of $x\$$.
2.  For any node $u$, let $P_u$ denote the set of strings that can be created by concatenating the labels of any path from the root to $u$. Subsequently, $P_u \subseteq Substr(x\$)$, and for any strings $p_1, p_2 \in P_u$, the set of ending positions of occurrences of $p_1$ and $p_2$ in $x\$$ are equivalent, i.e.,

$$\{j \mid p_1 = \hat{x}[j - |p_1| + 1..j]\} = \{j \mid p_2 = \hat{x}[j - |p_2| + 1..j]\}$$

    where $\hat{x} = x\$$. This implies that $P_u = \{p_u[1..|p_u|], \ldots, p_u[k..|p_u|]\}$ for some $p_u$ (the longest element of $P_u$) and $1 \leq k \leq |p_u|$.
3.  The suffix link $l(u)$ of a node $u$ points to a node $v$ such that the longest element of $P_v$ is $p'_u[2..|p'_u|]$, where $p'_u$ is the shortest element of $P_u$.

For technical convenience, we can also consider an auxiliary node that the suffix link of the root points to, which has outgoing edges for all symbols to the root. $D(x\$)$ can be built in $\mathcal{O}(m \log \sigma)$ time and $\mathcal{O}(m)$ space [16]. Because the suffix links form a tree, we can also process the suffix link tree in linear time so that each node $u$ holds a pointer $l'(u)$ to the deepest ancestor $v$ of $u$ (possibly $v = u$) that has $\$$ as an outgoing edge, i.e., the longest element $p_v$ of $P_v$ is the longest suffix of any $p \in P_u$ that is a suffix of $x$.

Using $D(x\$)$, we can incrementally compute for each $j = 1, \ldots, n$, the position in the DAWG, which corresponds to the longest suffix $T[i..j]$ of $T[1..j]$ that is a substring of $x$ in $O(n \log \sigma)$ overall time. Initially, $i = 1$, and we start at the root of $D(x\$)$. For each character $T[j]$ for $j = 1, \ldots, n$ and current node $u$, we traverse $D(x\$)$ by following the edge labelled $T[j]$ from $u$ or if that edge does not exist, try again after following the suffix link to $l(u)$. When the suffix link is traversed, $i$ is incremented, so that the length of $T[i..j-1]$ matches the length of the longest element in $P_{l(u)}$ ($-1$ for the auxiliary node, and 0 for the root node).

If, upon reading character $T[j^*]$, we arrive at a node $u$, then, $v = l'(u)$ points to the (possibly empty) longest suffix $T[i^*..j^*]$ of $T[1..j^*]$ that is a suffix of $x$, so we have detected suffixes of $x$ with lengths $0..j^* - i^* + 1$ starting at positions $i \in [i^*, j^*]$ and ending at position $j^*$ of $T$. In this manner, we can find occurrences of all suffixes of $x$ in $T$ in $\mathcal{O}(n \log \sigma)$ time.

### 4.2. Finding Occurrences of Y

To enumerate all the occurrences of $y$ in $T$, we preprocess $y$ in $\mathcal{O}(\ell - m)$ time and constant space using the Crochemore–Perrin preprocessing algorithm [17]. We can then find all occurrences of $y$ in $T$ in $\mathcal{O}(n)$ time and constant space.

### 4.3. Finding Occurrences of Prefixes of Z

To enumerate all occurrences of prefixes of $z$ in $T$ we construct the suffix tree [18–20] $S(z\$)$ with suffix links. Note that $S(z\$)$ contains exactly one node $z'$, such that $path(z') = z\$$. For each explicit node in $S(z\$)$, we add a special pointer to the closest ancestor that is on the path from the root to $z'$. This takes $\mathcal{O}(m \log \sigma)$ time and $\mathcal{O}(m)$ space. We start at the root of $S(z\$)$ and for $i = 1, 2, \ldots, n$ we follow the edge labelled $T[i]$ until we reach a node $u$ that has no outgoing edges labelled $T[i]$ available.

We then use the special pointer at node $u$ to find the closest ancestor $v$ of $u$ that is on the path to $z'$. $path(v)$ then yields the longest prefix of $z$ at position $i = 1$ of $T$. Furthermore, every prefix of $path(v)$ is also a prefix of $z$, so we have detected prefixes of $z$ with lengths $0..|path(v)|$ all starting at position $i$. To find the longest prefix of $z$ at position $i + 1$ of $T$, follow the suffix link of the node $u$ we ended on for input $T[i]$ and repeat this process. In this manner, we can enumerate all the occurrences of all prefixes of $z$ in $T$ in $\mathcal{O}(n \log \sigma)$ time.

### 4.4. Putting the Occurrences Together

We now determine which substrings in the batch $B$ are not unique in $T$. Notice that, for any $1 \leq i_1 < i_2 < i_3 \leq k$, occurrences of $B_{i_1}$ and $B_{i_3}$ which share the same occurrence of $y$ implies an occurrence of $B_{i_2}$ also sharing the occurrence of $y$. Therefore, we maintain an integer arrays $R$ of size $k + 1$, where all elements are initially 0, in order to record the start and end of ranges in $B$ that have been found to occur in $T$.

We use the DAWG $D(x\$)$, the Crochemore–Perrin algorithm for $y$, and the Suffix Tree $S(z\$)$, and maintain three parallel scans on $T$, shifted so that the three parts are detected in sync. More precisely, at position $i$, we maintain the following:

1. the longest suffix $x^*$ of $T[i - \ell - k + 2..i - \ell]$ that is also a suffix of $x$ using the modified DAWG $D(x\$)$,
2. whether $y = T[i - \ell + 1..i - k + 1]$ using the Crochemore–Perrin algorithm, and
3. the longest prefix $z^*$ of $T[i - k + 2..i]$ that is also a prefix of $z$ while using the modified suffix tree $S(z\$)$.

If $|x^*| + |y| + |z^*| < \ell$, then no substrings in $B$ occur here. On the other hand, if $|x^*| + |y| + |z^*| \geq \ell$, then certainly one or more substrings in $B$ occur here. Specifically, if $|x^*| + |y| + |z^*| \geq \ell$, then substrings $B_{k-|x^*|}$ through $B_{k+|y|+|z^*|-\ell}$ in $B$ occur here. We increment $R[k - |x^*|]$ and decrement $R[k + |y| + |z^*| - \ell + 1]$.

After the scan, we process $R$ and compute for $i = 1, \ldots, k$, $R'[i] = \sum_{j=1}^{i} R[j]$, i.e., $R'[1] = R[1]$ and $R'[i] = R'[i - 1] + R[i]$, which represents the total number of occurrences of $B_i$ in $T$ (including those which contain $q$). Thus, the total is $\mathcal{O}(n \log \sigma)$ time. At any moment $\mathcal{O}(m)$ space is used.

We have shown that we can determine the uniqueness of all the substrings in a batch $B$ in $T$ in $\mathcal{O}(n \log \sigma)$ time and $\mathcal{O}(m)$ space. Because there are $\lceil \frac{\ell}{m} \rceil$ batches of possible substrings of length $\ell$ covering $p$, we can determine the uniqueness of all substrings in all batches in $\mathcal{O}((\frac{\ell}{m} + 1)n \log \sigma)$ time and using at most $\mathcal{O}(m)$ words at any moment.

**Theorem 4.** *There is a deterministic algorithm that computes the shortest unique substring (SUS) of a text $T[1..n]$ that covers some query position $p$ chosen at runtime in $\mathcal{O}(n/\tau)$ words of workspace and $\mathcal{O}(n\tau \log \sigma \log n)$ time.*

**Proof of Theorem 4.** If there is a unique substring in $T$ with length $\ell$ and $\ell < n$, then there is a unique substring in $T$ with length greater than $\ell$. This property lets us use exponential search over the length $\ell$ of the shortest unique substring (SUS), with Lemma 1 as a sub-algorithm, in order to find the SUS that covers a query position $p$ in $\mathcal{O}(m)$ words and $\mathcal{O}((\frac{L}{m} + 1)n \log \sigma \log L)$ time, where $L$ is the length of the SUS. Setting $m = n/\tau$, this yields a time complexity of $\mathcal{O}((L\tau + n) \log \sigma \log L) \subseteq \mathcal{O}(n\tau \log \sigma \log n) \subset o(n\tau^2 \log \sigma \log(n/\tau))$. $\square$

Algorithm 2 shows the pseudo-code.

---

**Algorithm 2** An $O((\frac{L}{m} + 1)n \log \sigma \log L) \subseteq \mathcal{O}(n\tau \log \sigma \log n)$-time algorithm to find an SUS of $T[1..n]$ containing $T[q]$, where $L$ is the length of the SUS.

---

 1   for next choice of $\ell$ in binary search for $L \in [1, n]$
 2   % If there exists a unique substring of length $\ell$ that contains position $q$, then $L \leq \ell$.
 3   % Otherwise, all of them are repeating so $L > \ell$
 4      for $i$ from $q - \ell + 1$ to $q$ step $m$
 5      % check all substrings of length $\ell$ that include $T[q]$ in batches of size $k$ ($m$ except possibly last)
 6         $k = min(m, q - i + 1)$
 7         $x = T[i..i + k - 2], y = T[i + k - 1..i + \ell - 1], z = T[i + \ell..i + \ell + k - 2]$
 8         Construct the DAWG $D(x\$)$
 9         Preprocess $y$ for the Crochemore-Perrin algorithm
10         Construct the suffix tree $S(z\$)$
11         for $j$ from 1 to $k + 1$
12            $R[j] = 0, R'[j] = 0$
13         end for
14         for $j$ from 1 to $n$
15            Compute longest common suffix $x^*$ of $x$ and $T[j - \ell - k + 2..j - \ell]$ using $D(x\$)$
16            Compute whether $y = T[j - \ell + 1..j - k + 1]$, using Crochemore-Perrin algorithm
17            Compute longest common prefix $y^*$ of $y$ and $T[j - k + 2..j]$ using $S(z\$)$
18            if $|x^*| + |y| + |z| \geq \ell$ then
19               $R[k - |x^*|] = R[k - |x^*|] + 1$
20               $R[k + |y| + |z^*| - \ell + 1] = R[k + |y| + |z^*| - \ell + 1] - 1$
21            end if
22         end for
23         $R'[1] = R[1]$
24         for $j$ from 2 to $k$
25            $R'[j] = R'[j - 1] + R[j]$
26         end for
27         % $T[i + j - 1..i + j + \ell - 2]$ is unique if and only if $R'[j] = 1$
28      end for
29  end for

---

## 5. Future Work

It seems straightforward to generalize our algorithms to find an SUS containing a given interval instead of a single position, or finding substrings whose Parikh vectors are unique, but it might be more challenging to generalize them to finding substrings whose elements' relative order is unique, or to higher dimensions. Suppose that we want to find the smallest $d$-dimensional hypercube containing a given cell in a $d$-dimensional matrix, which occurs only once in that matrix. If the side lengths of the hypercube and matrix are $L$ and $n$, respectively, then our approach seems to slow down by a factor of $(Ln)^{d-1}$. However, if we instead want to find the smallest $d$-dimensional hyperrectangle, then the slowdown seems to be more like $L^{2d}n^d$.

**Author Contributions:** Conceptualization, G.H.; Formal analysis, H.B., T.G., G.H., S.J.P., L.M.S.R.; Investigation, H.B., T.G., G.H., S.J.P., L.M.S.R.; Writing—H.B., T.G.; Writing—review and editing, H.B., T.G., G.H., S.J.P., L.M.S.R.; Project administration, T.G. All authors have read and agreed to the published version of the manuscript.

**Funding:** H.B. was partially funded by JSPS KAKENHI Grant Numbers JP16H02783, JP20H04141. T.G. was partially funded by NSERC grant RGPIN-07185-2020. S.J.P. was partially supported by Academy of Finland grant 319454. L.M.S.R. was supported by national funds through Fundação para a Ciência e a Tecnologia (FCT) with reference UIDB/50021/2020. This paper resulted from a meeting at INESC in 2017 funded by the EU's Horizon 2020 research and innovation programme under Marie Skłodowska-Curie grant agreement No 690941 (BIRDS).

**Acknowledgments:** The authors are very grateful to Tatiana Starikovskaya and Sharma Thankachan for helpful discussions. The second author is also grateful to Starikovskaya for once saving him from a wild boar.

**Conflicts of Interest:** The authors declare no conflict of interest.

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
