# Peer review of "More Time-Space Tradeoffs for Finding a Shortest Unique Substring"

_algorithms, doi:10.3390/a13090234_

Round 1
Reviewer 1 Report
The paper presents new results on finding shortest unique substrings. The reasoning builds heavily on earlier results combing them in a clever way.
The paper is not self-contained and it has been written only for experts of the topic. The content is concise without any examples although there is no strict space limit in open-access publishing.
REMARKS
- Your definition of SUS is not precise. I had to look at earlier papers for an exact definition.
- "High probability" in theorems should be specified.
SMALL THINGS
- Line 12: in T
- Fig. 1, line 3: $q$
- Ref. [9] is an M.Sc. thesis.
Author Response
> The paper is not self-contained and it has been written only for experts of the topic. The content is concise without any examples although there is no strict space limit in open-access publishing.
We have added more exposition and a link to Abedin et al.'s recent survey.
> Your definition of SUS is not precise. I had to look at earlier papers for an exact definition.
We have given a more precise definition and examples.
> "High probability" in theorems should be specified.
We have added a restriction that allows us to make "with high probability" mean "the probability of failure is 1 over any given polynomial of n".
> Line 12: in T
Fixed.
> Fig. 1, line 3: $q$
Fixed.
> Ref. [9] is an M.Sc. thesis.
Fixed.
Reviewer 2 Report
please see the attachment

Author Response
> The significance and method of this study are not clearly explained in abstract.
Explain specifically, pls.
We have extended the abstract and tried to make it more informative.
> Generally, when citing references, advice cite them respectively in introduction.
We have included all the citations in the introduction.
Reviewer 3 Report
The paper presents new time-space tradeoffs for the problem of finding shortest unique substring (SUSs) and approximate (k-mismatch) SUSs in a text T, of length n, with alphabet size \sigma.
The authors extend previous results by Senanayaka [9] to find, with high probability, an SUS of length L in O((L/m)n \log L) time using random access over T.
The algorithm is improved by replacing the Karp-Rabin pattern matching used in [9] by a Monte-Carlo randomized sketching method, presented in [13]. The resulting running time is O((L/m)n \log^2(L) \log \log \sigma) taking O((L/m)\log^2 L) sequential passes over T.
The authors also show how to find approximate (k-mismatch) SUSs using results by Gawrychowski and Starikovskaya [14] in O(n^{1+\epsilon}L/m) time and O(n^{\epsilon}L/m) sequential scans over T, for some positive \epsilon and constant k.
In the second part, the paper presents a deterministic algorithm based on DAWG searches [15] for finding (exact) SUSs in O(n\tau \log \sigma \log n) time using O(n/\tau) words of extra space, which improves the best result by Ganguly et al. [8].
There are interesting insights in the paper, the results are sound, and the topic is interesting with applications in Bioinformatics.
I suggest to accept the paper.
########
The authors could improve some parts of the manuscript as following.
Major:
1. The background is too short, it requires a good previous knowledge from the reader. I recommend to indicate a textbook for the prerequisites.
2. A table with the summary of results and their theoretical bounds, discussed in Section 1, would be welcome.
3. A pseudo-code (like in Figure 1) for the algorithm presented in Section 4 would be good.
########
Minor:
- p2l52: "suffix tree" --> "suffix tree [17-19]" (include citation)
- p2l76: "of the \ell substrings of" --> "of the substrings of"
- p2l77: "lg n" --> "log_2 n"
- p4l129: "qrs" ??
- p4l135: "labeles of" --> "labels of"
- p5l165: "i = [1,2,...,n]" --> "i = 1..n"
- p6l178: "use the the" --> "use the"
Author Response
> 1. The background is too short, it requires a good previous knowledge from the reader. I recommend to indicate a textbook for the prerequisites.
We have extended the introduction and cited Abedin et al.'s recent survey of results on SUSs.
> 2. A table with the summary of results and their theoretical bounds, discussed in Section 1, would be welcome.
Added.
> A pseudo-code (like in Figure 1) for the algorithm presented in Section 4 would be good.
Added.
> p2l52: "suffix tree" --> "suffix tree [17-19]" (include citation)
Fixed.
> p2l76: "of the \ell substrings of" --> "of the substrings of"
Fixed.
> p2l77: "lg n" --> "log_2 n"
Fixed.
> p4l129: "qrs" ??
Fixed (changed to "q r s").
> p4l135: "labeles of" --> "labels of"
Fixed.
> p5l165: "i = [1,2,...,n]" --> "i = 1..n"
Fixed.
> p6l178: "use the the" --> "use the"
Fixed.